# Intra-Modal Neighbors Never Lie: Rectifying Inter-Modal Noisy Correspondence via Graph-Based Intra-Modal Reasoning

Yang Liu [1 2]  Wentao Feng [1 2]  Shu-Dong Huang [1 2]  Yalan Ye [3]  Jiancheng Lv [1 2]

## Abstract

Large-scale web-harvested datasets have fueled the progress of cross-modal retrieval but inevitably suffer from *noisy correspondence*, which severely degrades model generalization. Existing methods primarily address this by filtering out noise or seeking a substitute label, yet they predominantly remain bound by a "Discrete Selection" paradigm. We argue that relying on a single discrete proxy induces *Single-Point Fragility* and *Discretization Error*. To overcome these limitations, we propose a novel framework, **Intra-modal Neighbor-aware Noise Rectification (IN$^2$R)**, which shifts the paradigm from searching for a substitute to *synthesizing* a reliable supervision target. Leveraging the intrinsic geometric stability of intra-modal data, IN$^2$R employs a **Graph Refiner** to perform relational reasoning over neighbors retrieved from a dynamic **Cross-Model Memory**. Instead of propagating discrete labels, our method synthesizes a continuous, soft prototype that reflects the consensus of the local semantic neighborhood, effectively rectifying inter-modal misalignment. Extensive experiments on Flickr30K, MS-COCO, and CC152K demonstrate that IN$^2$R significantly outperforms state-of-the-art methods. Our code and pretrained models are publicly available at https://github.com/liuyyy111/IN2R.

## 1. Introduction

Effective visual-semantic alignment has emerged as a cornerstone of diverse vision-language applications, encompass-

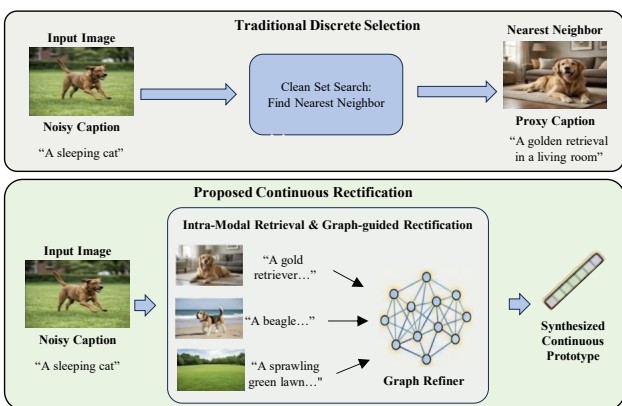

*Figure 1.* Comparison between the Traditional Discrete Selection paradigm and our proposed Continuous Rectification (IN$^2$R). While discrete selection (top) seeks a single substitute proxy from a finite dataset, often suffering from discretization error or selecting noisy neighbors (e.g., retrieving an imperfect caption), our approach (bottom) leverages the intrinsic topological structure. By retrieving intra-modal neighbors and aggregating them via a Graph Refiner, we synthesize a robust, continuous prototype that rectifies the semantic misalignment (e.g., correcting "A sleeping cat" using the visual consensus of dog-related features).

ing cross-modal retrieval, visual question answering, and broader multi-modal reasoning. The prevailing paradigm relies on contrastive learning to align visual and textual representations into a shared semantic space, typically requiring large-scale, high-quality image-text pairs. However, real-world datasets derived from web sources inevitably suffer from noisy correspondence, leading to images being paired with mismatched or irrelevant captions. Recent studies have revealed that even widely used benchmarks like Conceptual Captions (Huang et al., 2021) contain a non-negligible ratio of mismatched pairs. Such noise fundamentally corrupts the training signal, causing the model to memorize erroneous associations and severely degrading retrieval performance.

To combat this, existing approaches primarily diverge into three paradigms. Sample Selection methods (Huang et al., 2021; Qin et al., 2022) and Consistency-based approaches (Yang et al., 2023; Zha et al., 2025; Zhao et al., 2024) often falter due to intrinsic data waste (i.e., discarding informative hard positives) or a circular dependency on corrupted inter-modal signals for re-weighting. Consequently, recent

[1]College of Computer Science, Sichuan University, China [2]Engineering Research Center of Machine Learning and Industry Intelligence, Ministry of Education [3]School of Computer Science and Engineering, University of Electronic Science and Technology of China, China. Correspondence to: Shu-Dong Huang <huangsd@scu.edu.cn>.

*Proceedings of the 43$^{rd}$ International Conference on Machine Learning*, Seoul, South Korea. PMLR 306, 2026. Copyright 2026 by the author(s).

research has shifted towards Correction-based strategies (Han et al., 2023; Li et al., 2024; Han et al., 2024), which attempt to rectify noisy labels by retrieving new targets. However, we argue these methods remain bound by a **"Discrete Selection"** paradigm, seeking a substitute proxy from the existing finite dataset. This reliance on discrete proxies fundamentally limits precision: it induces **Single-Point Fragility** when the selected neighbor is noisy, and inevitably introduces **Discretization Error** by forcing continuous semantic truths to align with imperfect, discrete samples.

To overcome the fragility of discrete selection, we exploit the **intrinsic topological structure** of the data. A key observation, illustrated in Figure 1, motivates our design: *while noise disrupts the explicit alignment between modalities, the implicit geometric structure within each modality remains robust*. For instance, as shown in the upper path of Figure 1, an image of a golden retriever might be wrongly paired with a noisy caption "a sleeping cat". Traditional methods risk selecting another discrete proxy that may itself be imperfect. However, the image still maintains correct semantic proximity to other dog images (e.g., "beagle") in the visual feature space. *This implies that reliable semantic information is not attached to any single instance—which might itself be noisy—but is effectively preserved within the collective consensus of the local neighborhood.* **Crucially, we posit that the semantic "truth" of a noisy sample is not a discrete point waiting to be found in the dataset, but is best modeled as a continuous prototype synthesized from this local consensus.** This perspective drives a paradigm shift from searching for a substitute to synthesizing a target, offering two decisive advantages. First, regarding **robustness**, it mitigates the risk of selecting a noisy neighbor; while individual samples may be unreliable, their collective distribution statistically marginalizes such noise. Second, regarding **precision**, it transcends the limitations of discrete selection. Unlike reusing a neighbor's existing label—which inevitably introduces discretization error and merely recycles old data—our method synthesizes a **new, continuous supervision signal**. This synthesized target fills the semantic gaps between discrete samples, providing **finer-grained supervision** that more accurately approximates the true semantic center of the manifold.

To instantiate this continuous rectification paradigm, we propose a novel framework termed Intra-modal Neighbor-aware Noise Rectification (IN$^2$R). Recognizing that reliable "synthesis" requires a pristine source of information, we build our framework upon a co-training backbone, employing dual peer networks that maintain Cross-Model Memory Queues to dynamically curate high-confidence clean samples. This cross-model design is pivotal: it decouples the source of retrieval from the model being trained, ensuring that the topological reference remains unbiased. The core innovation lies in our Graph-Guided Rectification mech-

anism. Rather than treating the retrieved neighbors as a bag of discrete candidates, we model them as a local semantic graph. Specifically, we employ a learnable Graph Refiner that performs relational reasoning over the retrieved neighbors. Through attention-based aggregation, the graph refiner captures the subtle geometric dependencies within the neighborhood and synthesizes a refined prototype. This prototype serves as a robust, continuous supervision target that is both topologically faithful (derived from the clean manifold) and statistically precise (marginalizing discrete noise), effectively transforming the supervision of noisy data from "imitation of a proxy" to "alignment with a synthesized truth."

The main contributions are summarized as follows:

- **Paradigm Shift:** We identify the "Single-Point Fragility" in discrete selection methods and propose a shift towards *continuous rectification*. Our approach leverages intra-modal topology to synthesize robust supervision targets, avoiding discretization errors.

- **Methodological Innovation:** We propose the IN$^2$R framework, which integrates a Cross-Model Memory with a Graph Refiner. This design effectively decouples noise sources and employs relational reasoning to generate fine-grained supervision for noisy samples.

- **State-of-the-Art Performance:** Extensive experiments on three benchmarks demonstrate that IN$^2$R significantly outperforms existing methods, particularly in high-noise scenarios, validating the efficacy of our strategy.

## 2. Related Works

**Image-Text Matching.** Classical image–text retrieval approaches can be broadly categorized into two groups based on the granularity of the matching similarity: global-level matching and local-level matching methods. Global-level matching methods project images and texts into a shared embedding space, where similarity is computed using improved loss functions (Faghri et al., 2018; Chun et al., 2021) to bring semantically aligned pairs closer together. To enhance the quality of the embedding space, recent studies have introduced more sophisticated network architectures, such as graph convolutional networks (Liu et al., 2020; Wang et al., 2020), generalized pooling operators (Chen et al., 2021), and other advanced model designs (Huang et al., 2018; Li et al., 2019; 2022). To achieve closer alignment between semantically corresponding image–text pairs, several methods (Chun et al., 2021; Kim et al., 2023; Chun, 2023; Liu et al., 2025b;a) have been proposed to address the inherent discrepancy in information density between visual and textual modalities. In contrast, local-level matching methods (Diao et al., 2021; Liu et al., 2023; Wang et al.,

2019; Wei et al., 2020; Zhang et al., 2022; 2020; Chen et al., 2020; Qu et al., 2021) focus on fine-grained region-level alignment. These approaches typically capture detailed correspondences between image regions and textual phrases through cross-modal interaction networks.

**Learning with Noisy Correspondence (NCL).** Existing NCL methods primarily diverge into three streams: selection, consistency, and correction. Sample Selection methods (Huang et al., 2021; Qin et al., 2022) partition data into clean and noisy subsets based on loss distributions. While effective for low noise, these methods inevitably suffer from data waste by discarding "hard positives" that exhibit high losses. To mitigate this, Consistency-based methods (Yang et al., 2023; Zha et al., 2025; Zhao et al., 2024) re-weight samples by enforcing cross-modal or intra-modal consistency. However, they face a circular dependency: consistency computed from corrupted inter-modal signals is inherently unreliable under high noise ratios. Recently, Correction-based approaches have emerged, attempting to rectify noisy labels by learning meta-similarity (Han et al., 2023), mining consistency cues across views (Ma et al., 2024), suppressing soft-margin contributions of suspicious pairs (Yang et al., 2024), retrieving nearest neighbors (Li et al., 2024), propagating pseudo-label consistency across peers (Liu et al., 2026), or seeking optimal transport plans (Han et al., 2024). Crucially, these methods predominantly follow a **"Discrete Selection"** paradigm—seeking a substitute proxy from the finite dataset. We argue this induces **discretization error**, as the true semantic target often lies on a continuous manifold and may not exist in the discrete candidate pool. Unlike these works, our method shifts from finding a discrete proxy to synthesizing a continuous prototype.

**Graph-based Reasoning in NCL**. Graph Neural Networks (GNNs) have recently been introduced to Noisy Correspondence Learning (NCL) to capture high-order structural information. Recent works like GLP (Li et al., 2025) and SPS (Xie et al., 2025) construct neighbor graphs to refine representations. However, these approaches typically treat the graph as a tool for **passive label propagation** or smoothing, averaging out discriminative signals alongside noise. In contrast, our framework employs GNNs for **active reasoning**, utilizing attention mechanisms to dynamically detect structural conflicts and synthesize robust supervision signals from the local consensus, rather than merely propagating existing labels.

## 3. Method

### 3.1. Overview and Problem Formulation

We consider the cross-modal retrieval task given a dataset $\mathcal{D} = \{(I_i, T_i)\}_{i=1}^{N}$, which inevitably contains noisy corre-

spondence. Our goal is to learn robust encoders $f_\theta(\cdot)$ and $g_\phi(\cdot)$ by transforming noisy labels into continuous, high-fidelity supervision signals.

We propose the **Intra-modal Neighbor-aware Noise Rectification ($\text{IN}^2\text{R}$)** framework. As illustrated in Figure 2, our method adopts a co-training paradigm with two peer networks, $\mathcal{M}_A$ and $\mathcal{M}_B$, to decouple the noise identification from the rectification process.

To initiate the robust training, we strictly leverage the "Small-Loss" hypothesis. Specifically, the training proceeds in two phases:

- **Warm-up Phase:** We first warm up the networks on the full dataset $\mathcal{D}$. Following L2RM (Han et al., 2024), we adopt the Symmetric Cross Entropy (SCE) as the robust objective to prevent overfitting. This strategy helps establish a preliminary discriminative feature space, effectively separating the loss distributions of clean and noisy samples.

  To formalize this, let $\mathbb{H}(\cdot, \cdot)$ denote the cross-entropy operator. We define the general SCE loss between a target distribution $\mathbf{q}$ and a prediction distribution $\mathbf{p}$ as:

  $$\mathcal{L}_{sce}(\mathbf{q}, \mathbf{p}) = \alpha \mathbb{H}(\mathbf{q}, \mathbf{p}) + \beta \mathbb{H}(\mathbf{p}, \mathbf{q}) \qquad (1)$$

  where the first term is the standard InfoNCE and the second is the Reverse Cross Entropy (RCE). Here, $\mathbf{p}$ denotes the temperature-scaled softmax distribution of similarity scores, while the target $\mathbf{q}$ is set to the one-hot ground-truth $\mathbf{y}$. Note that we apply $\epsilon$-smoothing to $\mathbf{q}$ in the RCE term for numerical stability. The bidirectional robust loss is derived as $\mathcal{L}_{robust} = \frac{1}{2}[\mathcal{L}_{sce}(\mathbf{y}, \mathbf{p}^{i2t}) + \mathcal{L}_{sce}(\mathbf{y}, \mathbf{p}^{t2i})]$.

- **Co-training Phase with Dynamic Partition:** Following warm-up, at each epoch, we fit a two-component Gaussian Mixture Model (GMM) to the per-sample loss distributions. Based on the posterior probabilities, we dynamically partition $\mathcal{D}$ into a labeled clean subset $\mathcal{D}_{clean}$ and an unlabeled noisy subset $\mathcal{D}_{noisy}$.

### 3.2. Manifold Stabilization via Intra-modal Constraints

For the identified clean subset $\mathcal{D}_{clean}$, our goal is twofold: (1) to align the semantic representations across modalities, and (2) to explicitly consolidate the geometric structure within each modality to support reliable neighbor retrieval.

Formally, consider a batch of clean samples $\mathcal{B} = \{(I_i, T_i)\}_{i=1}^{B}$ sampled from $\mathcal{D}_{clean}$. Let $\mathbf{v}_i = f_\theta(I_i)$ and $\mathbf{u}_i = g_\phi(T_i)$ denote the normalized image and text embeddings. We define the standard **bi-directional Triplet**

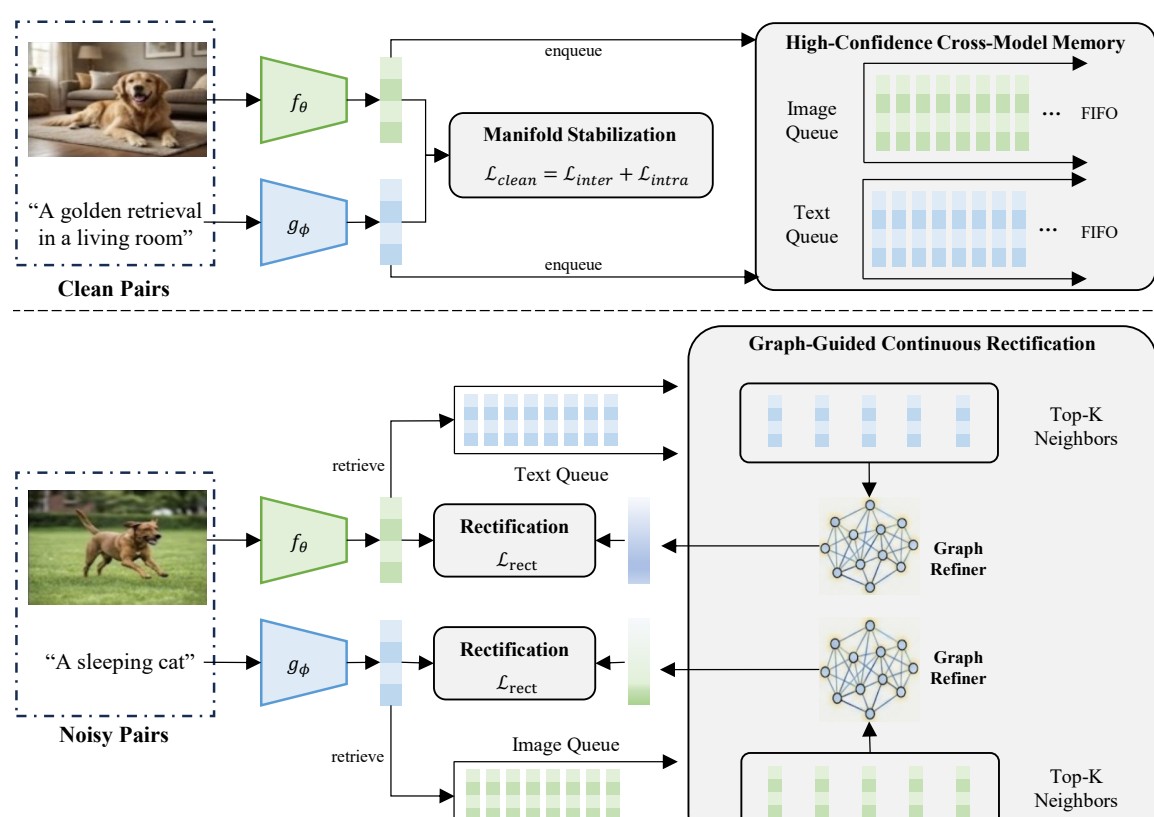

*Figure 2.* **The overall framework of Intra-modal Neighbor-aware Noise Rectification (IN$^2$R). (Top) Manifold Stabilization:** For identified clean pairs, we minimize $\mathcal{L}_{\text{clean}}$ (combining inter-modal alignment and intra-modal constraints) to consolidate the geometric structure, while pushing high-confidence representations into the *Cross-Model Memory*. **(Bottom) Graph-Guided Continuous Rectification:** For noisy pairs, we retrieve the Top-$K$ intra-modal neighbors from the memory queue. A learnable *Graph Refiner* then performs relational reasoning over these neighbors to synthesize a continuous, robust soft prototype. This synthesized target provides fine-grained supervision via $\mathcal{L}_{\text{rect}}$, correcting the noisy correspondence.

**Ranking Loss $\mathcal{L}_{triplet}$** as:

$$
\mathcal{L}_{triplet}(\mathbf{X}, \mathbf{Y}) = \sum_{i=1}^{B} \Big( [\alpha - S(\mathbf{x}_i, \mathbf{y}_i) + S(\mathbf{x}_i, \mathbf{y}_i^-)]_+
$$
$$
+ [\alpha - S(\mathbf{x}_i, \mathbf{y}_i) + S(\mathbf{x}_j^-, \mathbf{y}_i)]_+ \Big)
\tag{2}
$$

where $S(\cdot, \cdot)$ computes the cosine similarity, $\alpha$ is the margin parameter, $[x]_+ = \max(0, x)$, and $\mathbf{y}_i^- = \arg\max_{j \neq i} S(\mathbf{x}_i, \mathbf{y}_j)$ and $\mathbf{x}_j^- = \arg\max_{j \neq i} S(\mathbf{x}_j, \mathbf{y}_i)$ denote the hardest negatives within the batch.

### 3.2.1. INTER-MODAL ALIGNMENT

To bridge the modality gap and align semantic semantics, we apply the ranking loss to the paired image and text embeddings:

$$
\mathcal{L}_{inter} = \mathcal{L}_{triplet}(\mathbf{V}, \mathbf{U})
\tag{3}
$$

where $\mathbf{V}$ and $\mathbf{U}$ represent the sets of embeddings in the batch.

### 3.2.2. INTRA-MODAL GEOMETRIC CONSISTENCY

Relying solely on inter-modal alignment often leads to semantic misalignment within individual modalities, where semantically similar instances may not be clustered effectively. Inspired by ConVSE (Liu et al., 2022), we introduce explicit intra-modal constraints to regularize the feature space.

Following the strategy in (Liu et al., 2022), we employ **Random Dropout** as a data augmentation technique to generate positive pairs without requiring external data. Specifically, we forward the same batch of images (or texts) through the encoder twice with different dropout masks, yielding two views for each sample: original views $\{\mathbf{v}_i, \mathbf{u}_i\}$ and augmented views $\{\mathbf{v}_i', \mathbf{u}_i'\}$.

We then impose the ranking constraints *within* each modality to pull these intrinsic positive pairs together while pushing

away different instances:

$$\mathcal{L}_{img} = \mathcal{L}_{triplet}(\mathbf{V}, \mathbf{V}'), \quad \mathcal{L}_{txt} = \mathcal{L}_{triplet}(\mathbf{U}, \mathbf{U}') \quad (4)$$

By explicitly enforcing $\mathcal{L}_{img}$ and $\mathcal{L}_{txt}$, we ensure that the visual (or textual) proximity faithfully reflects semantic similarity, preventing the manifold from collapsing due to noise.

### 3.2.3. OPTIMIZATION OBJECTIVE FOR CLEAN DATA

The final objective for the clean subset integrates both constraints:

$$\mathcal{L}_{clean} = \mathcal{L}_{inter} + \lambda_{intra}(\mathcal{L}_{img} + \mathcal{L}_{txt}) \quad (5)$$

where $\lambda_{intra}$ balances the contribution of the manifold regularization. This structure-first approach ensures a robust topological backbone for the subsequent rectification of noisy data.

### 3.3. High-Confidence Cross-Model Memory

To rectify noisy samples, we require a pristine source of semantic reference. Relying solely on the current minibatch is insufficient due to its limited scope, while using a static dataset fails to track the evolving feature space. Therefore, we maintain a **Dynamic Cross-Model Memory Bank** $\mathcal{Q}$ to store a history of reliable representations.

### 3.3.1. ELITIST ROLLING UPDATE STRATEGY

Standard memory banks typically enqueue all clean samples. However, the identified clean set $\mathcal{D}_{clean}$ may still contain "hard" noise with borderline confidence. Moreover, as the encoders evolve, stored features from early epochs become stale. To address these issues, we employ an **Elitist Rolling Update** mechanism:

- **Dynamic High-Confidence Filtering:** We compute a dynamic threshold $\tau_{dyn}^{(t)}$ at epoch $t$ as the average confidence of the current clean set. Only samples satisfying $p_i > \tau_{dyn}^{(t)}$ are considered as elite candidates.

- **FIFO Maintenance with Gradient Detachment:** We maintain the memory as a First-In-First-Out (FIFO) queue. In each iteration, the embeddings of these elite candidates are *detached* from the computation graph and pushed into the queue, replacing the oldest entries.

This ensures that the memory bank $\mathcal{Q}$ preserves only the most trustworthy and up-to-date representations of the manifold.

### 3.3.2. CROSS-MODEL DECOUPLING

To prevent *confirmation bias*—where a network reinforces its own erroneous predictions—we leverage the co-training

architecture to decouple the retrieval source from the query. Specifically, network $\mathcal{M}_A$ retrieves neighbors exclusively from the memory queue of its peer $\mathcal{M}_B$, denoted as $\mathcal{Q}_B$, and vice versa:

$$\mathcal{Q}_B = \{(\mathbf{v}_k, \mathbf{u}_k)\}_{k=1}^{M} \quad (6)$$

where $M$ is the memory capacity. By querying the peer's historical consensus, the model avoids verifying its own potential hallucinations.

### 3.4. Graph-Guided Continuous Rectification

For the identified noisy subset $\mathcal{D}_{noisy}$, the original labels are deemed unreliable. We propose to rectify them by synthesizing continuous supervision signals derived from the clean manifold. **Remark on Symmetry:** Our rectification process is designed to be **bidirectional and symmetric**. For a noisy pair $(I_i, T_i)$, we simultaneously rectify the image-to-text direction (synthesizing a **Soft Textual Prototype $\hat{\mathbf{t}}$**) and the text-to-image direction (synthesizing a **Soft Visual Prototype $\hat{\mathbf{v}}$**). For brevity, we detail the generation of $\hat{\mathbf{t}}$ given a noisy image query $\mathbf{q}_v = f_\theta(I_i)$; the generation of $\hat{\mathbf{v}}$ from a noisy text query $\mathbf{q}_u = g_\phi(T_i)$ follows an identical symmetric procedure.

### 3.4.1. INTRA-MODAL NEIGHBOR RETRIEVAL

We first query the peer memory bank $\mathcal{Q}_B$ to identify the local semantic support set. Specifically, we retrieve the Top-$K$ nearest neighbors based on the cosine similarity between the query $\mathbf{q}_v$ and the stored visual keys $\{\mathbf{v}_k\} \in \mathcal{Q}_B$. Crucially, to bridge the modality gap, we access the *paired textual values* of these neighbors, denoted as $\mathcal{U}_{neighbor} = \{\mathbf{u}_1, \dots, \mathbf{u}_K\}$, to serve as the candidate semantic targets.

### 3.4.2. PROTOTYPE SYNTHESIS VIA GRAPH REFINER

A naive approach would be to average $\mathcal{U}_{neighbor}$ (mean pooling) or select the top-1 candidate (discrete selection). However, the retrieved neighborhood may contain outliers or irrelevant samples. To mitigate this, we treat $\mathcal{U}_{neighbor}$ as nodes in a fully connected graph and employ a learnable **Graph Refiner** to synthesize a robust prototype.

The Graph Refiner consists of a Multi-Head Self-Attention (MHSA) layer followed by a feed-forward aggregation. Let $\mathbf{H} \in \mathbb{R}^{K \times d}$ denote the stacked features of $\mathcal{U}_{neighbor}$. We first compute the intra-neighborhood attention to model the relational density:

$$\text{Attn}(\mathbf{H}) = \text{Softmax}\left(\frac{(\mathbf{H}\mathbf{W}_Q)(\mathbf{H}\mathbf{W}_K)^T}{\sqrt{d_k}}\right)(\mathbf{H}\mathbf{W}_V) \quad (7)$$

where $\mathbf{W}_Q, \mathbf{W}_K, \mathbf{W}_V$ are learnable projection matrices. This mechanism assigns higher weights to neighbors that

*Table 1.* Image-Text Retrieval on Flickr30K and MS-COCO 1K datasets under different noise ratios. * indicates the noise robust method. The best indicators are represented in **bold**.

| Noise | Method | Flickr30k 1K Test | | | | | | | MS-COCO 5-fold 1K Test | | | | | | |
|---|---|---|---|---|---|---|---|---|---|---|---|---|---|---|---|
| | | Text Retrieval | | | Image Retrieval | | | rSum | Text Retrieval | | | Image Retrieval | | | rSum |
| | | R1 | R5 | R10 | R1 | R5 | R10 | | R1 | R5 | R10 | R1 | R5 | R10 | |
| 0.2 | NCR (NeurIPS'21) | 73.5 | 93.2 | 96.6 | 56.9 | 82.4 | 88.5 | 491.1 | 76.6 | 95.6 | 98.2 | 60.5 | 88.8 | 95.0 | 515.0 |
| | BiCro (CVPR'23) | 78.1 | 94.4 | 97.5 | 60.4 | 84.4 | 89.9 | 504.7 | 78.8 | 96.1 | 98.6 | 63.7 | 90.3 | 95.7 | 523.2 |
| | L2RM (CVPR'24) | 77.9 | 95.2 | 97.8 | 59.8 | 83.6 | 89.5 | 503.8 | 80.2 | 96.3 | 98.5 | 64.2 | 90.1 | 95.4 | 524.7 |
| | CREAM (TIP'24) | 77.4 | 95.0 | 97.5 | 58.7 | 84.1 | 89.8 | 502.3 | 78.9 | 96.3 | 98.7 | 63.3 | 90.5 | 95.3 | 523.0 |
| | ESC (CVPR'24) | 79.0 | 94.8 | 97.5 | 59.1 | 83.8 | 89.1 | 503.3 | 79.2 | 97.0 | 99.1 | 64.8 | 90.7 | 96.0 | 526.8 |
| | GSC (CVPR'24) | 78.3 | 94.6 | 97.8 | 60.1 | 84.5 | 90.5 | 505.8 | 79.5 | 96.4 | 98.9 | 64.4 | 90.6 | 95.9 | 525.7 |
| | SPS (IJCAI'25) | 79.5 | 95.0 | 98.0 | 60.5 | 84.3 | 89.8 | 507.1 | 79.8 | 96.4 | 98.6 | 64.3 | 90.5 | 95.8 | 525.5 |
| | PCSR (AAAI'26) | 78.7 | 95.1 | 97.9 | 60.9 | 83.7 | 89.4 | 505.7 | 80.5 | 97.3 | 98.9 | 63.8 | 89.7 | 95.1 | 525.4 |
| | **IN$^2$R** | 80.0 | 95.5 | 97.7 | 60.5 | 86.2 | 91.7 | 511.6 | 81.6 | 96.3 | 98.8 | 64.2 | 90.5 | 95.9 | 527.3 |
| 0.4 | NCR (NeurIPS'21) | 75.3 | 92.1 | 95.2 | 56.2 | 80.6 | 87.4 | 486.8 | 76.5 | 95.0 | 98.2 | 60.7 | 88.5 | 95.0 | 513.9 |
| | BiCro (CVPR'23) | 74.6 | 92.7 | 96.2 | 55.5 | 81.0 | 87.4 | 487.5 | 77.0 | 95.9 | 98.3 | 61.1 | 89.2 | 94.9 | 517.1 |
| | L2RM (CVPR'24) | 75.8 | 93.2 | 96.9 | 56.3 | 81.0 | 87.3 | 490.5 | 77.5 | 95.8 | 98.5 | 62.0 | 89.1 | 94.9 | 517.7 |
| | CREAM (TIP'24) | 76.3 | 93.3 | 97.1 | 57.0 | 82.6 | 88.7 | 495.0 | 76.5 | 95.0 | 98.2 | 61.7 | 89.4 | 95.6 | 516.8 |
| | ESC (CVPR'24) | 76.1 | 93.1 | 96.4 | 56.0 | 80.8 | 87.2 | 489.6 | 78.6 | 96.6 | 99.0 | 63.2 | 90.6 | 95.9 | 523.9 |
| | GSC (CVPR'24) | 76.5 | 94.1 | 97.6 | 57.5 | 82.7 | 88.9 | 497.3 | 78.2 | 95.9 | 98.2 | 62.5 | 89.7 | 95.4 | 519.9 |
| | SPS (IJCAI'25) | 77.8 | 93.6 | 97.1 | 57.3 | 83.5 | 89.6 | 498.9 | 79.2 | 95.9 | 98.5 | 63.3 | 89.8 | 95.4 | 522.1 |
| | PCSR (AAAI'26) | 76.6 | 94.5 | 97.3 | 57.4 | 82.4 | 88.9 | 497.1 | 77.7 | 96.0 | 98.1 | 63.1 | 89.4 | 95.5 | 519.8 |
| | **IN$^2$R** | 78.3 | 94.5 | 97.2 | 58.7 | 84.7 | 90.7 | 504.1 | 79.8 | 95.8 | 98.6 | 63.4 | 90.1 | 95.4 | 523.2 |
| 0.6 | NCR (NeurIPS'21) | 68.7 | 89.9 | 95.5 | 52.0 | 77.6 | 84.9 | 468.6 | 72.7 | 94.0 | 97.6 | 57.9 | 87.0 | 94.1 | 503.3 |
| | BiCro (CVPR'23) | 67.6 | 90.0 | 94.4 | 51.2 | 77.6 | 84.7 | 466.3 | 73.9 | 94.4 | 97.8 | 58.3 | 87.2 | 93.5 | 505.5 |
| | L2RM (CVPR'24) | 70.0 | 90.8 | 95.4 | 51.3 | 76.4 | 83.7 | 467.6 | 75.4 | 94.7 | 97.9 | 59.2 | 87.4 | 93.6 | 508.4 |
| | CREAM (TIP'24) | 70.6 | 91.2 | 96.1 | 53.3 | 79.2 | 87.0 | 477.4 | 74.7 | 94.7 | 98.0 | 59.7 | 88.0 | 94.6 | 509.9 |
| | ESC (CVPR'24) | 72.6 | 90.9 | 94.6 | 53.0 | 78.6 | 85.3 | 475.0 | 77.2 | 95.1 | 98.1 | 61.1 | 88.6 | 94.9 | 515.0 |
| | GSC (CVPR'24) | 70.8 | 91.1 | 95.9 | 53.6 | 79.8 | 86.8 | 478.0 | 75.6 | 95.1 | 98.0 | 60.0 | 88.3 | 94.6 | 511.7 |
| | SPS (IJCAI'25) | 73.4 | 92.7 | 96.3 | 53.7 | 80.2 | 87.7 | 484.1 | 77.6 | 95.7 | 98.3 | 61.6 | 89.0 | 95.1 | 517.2 |
| | PCSR (AAAI'26) | 72.8 | 91.9 | 95.8 | 54.8 | 79.7 | 86.5 | 480.5 | 76.3 | 94.8 | 97.9 | 61.7 | 87.5 | 94.0 | 512.2 |
| | **IN$^2$R** | 75.1 | 94.0 | 97.2 | 56.5 | 82.9 | 89.5 | 495.2 | 78.8 | 95.2 | 98.3 | 61.4 | 89.2 | 95.1 | 518.0 |
| 0.8 | NCR (NeurIPS'21) | 1.5 | 6.2 | 9.9 | 0.3 | 1.0 | 2.1 | 21.0 | 0.1 | 0.3 | 0.4 | 0.1 | 0.5 | 1.0 | 2.4 |
| | BiCro (CVPR'23) | 2.3 | 9.2 | 17.2 | 2.6 | 10.2 | 16.8 | 58.3 | 62.2 | 88.6 | 94.6 | 47.4 | 79.2 | 88.5 | 460.5 |
| | L2RM (CVPR'24) | 55.7 | 80.8 | 87.8 | 39.4 | 65.4 | 74.9 | 404.0 | 69.0 | 91.9 | 96.4 | 52.6 | 82.4 | 90.3 | 482.6 |
| | CREAM (TIP'24) | 58.7 | 83.3 | 90.1 | 40.8 | 67.1 | 76.3 | 416.3 | 68.6 | 92.0 | 96.4 | 52.4 | 84.8 | 92.8 | 487.2 |
| | PCSR (AAAI'26) | 63.1 | 87.1 | 93.2 | 44.6 | 70.9 | 78.6 | 437.5 | 71.6 | 92.9 | 96.7 | 55.3 | 83.9 | 93.3 | 493.7 |
| | **IN$^2$R** | 68.5 | 88.1 | 93.0 | 48.2 | 76.4 | 84.7 | 458.8 | 73.2 | 93.8 | 97.4 | 57.1 | 86.4 | 93.4 | 501.3 |

form a semantic consensus and suppresses outliers. The refined features are then obtained via a residual connection and layer normalization:

$$\mathbf{H'} = \text{LayerNorm}(\mathbf{H} + \text{Dropout}(\text{Linear}(\text{Attn}(\mathbf{H})))) \quad (8)$$

Finally, we perform mean pooling over the refined nodes $\mathbf{H'}$ to synthesize the continuous **Soft Textual Prototype** $\hat{\mathbf{t}}$:

$$\hat{\mathbf{t}} = \frac{1}{K} \sum_{k=1}^{K} \mathbf{H'}_k \quad (9)$$

Symmetrically, for a noisy text query, we obtain the **Soft Visual Prototype** $\hat{\mathbf{v}}$ using the same Graph Refiner shared across modalities.

### 3.4.3. RECTIFICATION OBJECTIVE

To utilize the embedding prototype $\hat{t}$ for supervision, we convert it into a **soft target distribution** $q^{i2t}(\hat{t})$ by computing its softmax-normalized similarity against the current batch embeddings $U$:

$$q^{i2t}(\hat{t}) = \text{Softmax}\left(\frac{\hat{t} \cdot U^\top}{\tau}\right) \quad (10)$$

Symmetrically, we derive $q^{t2i}(\hat{v})$ for the text-to-image direction. We then employ the robust objective from Eq. (1), utilizing these calibrated distributions as targets:

$$\mathcal{L}_{rect} = \frac{1}{2} \left[ \mathcal{L}_{sce}(q^{i2t}(\hat{t}), \mathbf{p}^{i2t}) + \mathcal{L}_{sce}(q^{t2i}(\hat{v}), \mathbf{p}^{t2i}) \right] \quad (11)$$

This objective aligns noisy samples with the intra-modal consensus while regularizing against estimation bias via the RCE term.

### 3.5. Overall Optimization

The final training objective integrates the structural constraints from clean data and the rectified supervision from noisy data. For each network (e.g., $\mathcal{M}_A$), the total loss is defined as a weighted sum:

$$\mathcal{L}_{total} = \mathcal{L}_{clean} + \gamma \mathcal{L}_{rect} \quad (12)$$

where $\mathcal{L}_{clean}$ (Eq. 5) consolidates the manifold structure using the hard-margin ranking loss, and $\mathcal{L}_{rect}$ (Eq. 11) guides the rectification using the robust symmetric loss. The hyper-parameter $\gamma$ balances the contribution of the synthesized supervision.

*Table 2.* Comparisons with real-world NCs on CC152K. The best performance is highlighted in **bold**.

| Method | Text Retrieval | | | Image Retrieval | | | rSum |
|---|---|---|---|---|---|---|---|
| | R@1 | R@5 | R@10 | R@1 | R@5 | R@10 | |
| NCR | 39.5 | 64.5 | 73.5 | 40.3 | 64.6 | 73.2 | 355.6 |
| BiCro | 40.8 | 67.2 | 76.1 | 42.1 | 67.6 | 76.4 | 370.2 |
| PC$^2$ | 39.3 | 66.4 | 75.4 | 39.8 | 66.4 | 76.8 | 364.1 |
| L2RM | 43.0 | 67.5 | 75.7 | 42.8 | 68.0 | 77.2 | 374.2 |
| ESC | 42.8 | 67.3 | 76.9 | 44.8 | 68.2 | 75.9 | 375.9 |
| GSC | 42.1 | 68.4 | 77.7 | 42.2 | 67.6 | 77.1 | 375.1 |
| SPS | 40.8 | 67.9 | 77.7 | 42.4 | 69.5 | 78.0 | 376.3 |
| PCSR | 43.7 | 67.7 | 77.2 | 43.1 | 67.7 | 76.3 | 375.3 |
| **IN$^2$R** | 45.2 | 69.0 | 76.8 | 43.3 | 68.3 | 78.2 | 380.8 |

# 4. Experiments

## 4.1. Datasets

We evaluate on three datasets: **Flickr30K** (Plummer et al., 2015), **MS-COCO** (Lin et al., 2014), and **Conceptual Captions (CC)** (Huang et al., 2021). **Flickr30K** contains 31K images (5 captions each); following (Faghri et al., 2018), we use 1K images each for validation and testing. **MS-COCO** comprises 123K images (5 captions each); we utilize 566K pairs for training, with 25K pairs each allocated for validation and testing. **CC** is a noisy, web-harvested dataset (1 caption each). We use the CC152K subset, consisting of 150K training images and 1K each for validation and testing.

**Noise Simulation and Evaluation.** Following (Huang et al., 2021), we assess retrieval performance using Recall at K (R@K), which quantifies the percentage of relevant items correctly identified within the top $K$ retrieved results. We report $R@1$, $R@5$, $R@10$, and the cumulative recall score (rSum) for bidirectional matching tasks.

## 4.2. Implementation Details

Following standard protocols in noisy correspondence learning (Huang et al., 2021; Han et al., 2024) we utilize pre-extracted features to ensure a fair comparison. For images, we use the bottom-up attention features extracted from a pre-trained Faster R-CNN (2048-d). For text, we use a Bi-GRU to extract sentence embeddings. These features are projected into a common $D$-dimensional metric space (e.g., $D = 1024$) via the learnable encoders $f_\theta$ and $g_\phi$. *More implementation details are provided in the supplementary material.*

## 4.3. Main Results

In this section, we carry out a comprehensive evaluation to present the effectiveness of IN$^2$R, benchmarking it against state-of-the-art (SOTA) baselines across three widely-used

datasets. The baselines comprise NCR (Huang et al., 2021), BiCro (Yang et al., 2023), L2RM (Han et al., 2024), CREAM (Ma et al., 2024), ESC (Yang et al., 2024), GSC (Zhao et al., 2024), SPS (Xie et al., 2025), and PCSR (Liu et al., 2026). We evaluate IN$^2$R on Flickr30K and MS-COCO with simulated noise rates of 20%–80% generated by random shuffling. Additionally, we validate performance on the real-world CC152K dataset, which contains inherent web noise. All reported test results are based on the optimal validation checkpoint.

**Evaluation under Simulated Noise.** Table 1 presents the comprehensive comparison between our proposed IN$^2$R and state-of-the-art methods on Flickr30K and MS-COCO datasets under varying symmetric noise ratios (20% to 80%). The results demonstrate that IN$^2$R consistently outperforms existing baselines across all noise levels and metrics.

*Robustness in High-Noise Regimes:* IN$^2$R excels as noise increases. At 80% noise, where NCR collapses (21.0 rSum), our method maintains remarkable stability. On Flickr30K, IN$^2$R achieves **458.8** rSum, surpassing PCSR by **21.3** points. Similarly, on MS-COCO, it sets a new SOTA of **501.3** (+7.6 points). This validates that our Graph-Guided Continuous Rectification effectively synthesizes reliable supervision even under extreme corruption.

*Improvements in Low-Noise Regimes:* Even at lower noise ratios (20% and 40%), where the clean data signal is stronger, IN$^2$R continues to refine the boundary of performance. For instance, at 20% noise on Flickr30K, our method achieves an rSum of **511.6**, outperforming the correction-based method SPS (507.1). This indicates that our manifold stabilization strategy and intra-modal reasoning are beneficial not just for correcting errors, but for learning a more discriminative feature space overall.

**Evaluation under Real-World Noise.**

We evaluate on Conceptual Captions (CC152K) to verify robustness against heterogeneous web noise. IN$^2$R generalizes remarkably well, achieving a state-of-the-art rSum of **380.8**, surpassing SPS (376.3) by **4.5** points. Notably, it attains a Text Retrieval R@1 of **45.2%**, significantly outperforming PCSR (43.7%). While baselines like ESC show isolated strengths, IN$^2$R delivers the most balanced performance across modalities. This confirms that our intra-modal rectification effectively handles naturally occurring mismatches without requiring manual noise priors.

## 4.4. Ablation Studies

To provide a comprehensive understanding of the proposed framework, we conduct extensive ablation studies on the Flickr30K dataset. Unless otherwise specified, we report results under the 60% symmetric noise setting, as high-noise scenarios best highlight the robustness of our rectification

mechanism.

**Impact of Key Components.** We investigate the contribution of each module in IN$^2$R by incrementally adding them to the baseline on the Flickr30K dataset under $60\%$ noise. The baseline, trained solely with the inter-modal ranking loss $\mathcal{L}_{inter}$, yields an rSum of 469.1. This relatively low performance highlights the difficulty of learning robust representations from heavily corrupted data without explicit correction.

Incorporating the intra-modal geometric constraints ($\mathcal{L}_{intra}$) improves the rSum to 475.2, a gain of 6.1 points. This suggests that stabilizing the feature manifold prevents the model from overfitting to noisy correspondence, providing a better initialization for retrieval. Furthermore, applying the Graph-Guided Rectification ($\mathcal{L}_{rect}$) directly to the baseline yields a significant boost, reaching 483.7 rSum. Finally, the full IN$^2$R framework, which integrates both manifold stabilization and continuous rectification, achieves the best performance with an rSum of **495.2**. This represents a substantial total improvement of **26.1** points over the baseline, confirming that the two components work synergistically: $\mathcal{L}_{intra}$ constructs a reliable geometric basis, while $\mathcal{L}_{rect}$ actively synthesizes precise supervision signals from it.

*Table 3.* **Component-wise ablation study** on Flickr30K ($60\%$ noise). $\mathcal{L}_{inter}$: Inter-modal alignment; $\mathcal{L}_{intra}$: Intra-modal geometric constraints; $\mathcal{L}_{rect}$: Graph-guided rectification.

| Components | | | Text Retrieval | | Image Retrieval | | | |
|---|---|---|---|---|---|---|---|---|
| $\mathcal{L}_{inter}$ | $\mathcal{L}_{intra}$ | $\mathcal{L}_{rect}$ | R@1 | R@10 | R@1 | R@10 | rSum | $\Delta$ |
| ✓ | | | 68.7 | 95.5 | 52.4 | 84.7 | 469.1 | - |
| ✓ | ✓ | | 71.2 | 96.1 | 52.8 | 85.3 | 475.2 | +[6.1] |
| ✓ | | ✓ | 73.6 | 96.8 | 53.7 | 86.8 | 483.7 | +[14.6] |
| ✓ | ✓ | ✓ | **75.1** | **97.2** | **56.5** | **89.5** | **495.2** | +[26.1] |

**Continuous Rectification vs. Discrete Selection.** To validate that continuous synthesis outperforms discrete selection, we compare our Graph Refiner with three strategies on Flickr30K ($60\%$ noise): **None** (discarding noise), **Hard Selection** (Top-1 neighbor), and **Mean Pooling** (Top-$K$ average). As shown in Table 4, *Hard Selection* (479.6) suffers from *Single-Point Fragility*, while *Mean Pooling* (485.2) fails to filter outliers. In contrast, our **Graph Refiner** achieves an rSum of **495.2**. By synthesizing a robust soft prototype via dynamic re-weighting, it outperforms discrete selection and mean pooling by **15.6** and **10.0** points, respectively.

**Impact of Refiner Architecture and Memory Decoupling.** We validate our architectural choices on Flickr30K ($60\%$ noise) by comparing our Multi-Head Self-Attention (MHSA) refiner against GCN and GAT variants. As shown in Table 5, MHSA achieves the best rSum of **495.2**, significantly outperforming GCN (486.3) and GAT (490.5). This superiority suggests that MHSA's dense, fully-connected

*Table 4.* **Comparison of different rectification strategies** on Flickr30K ($60\%$ noise). Our Graph Refiner (Continuous) outperforms Discrete Selection (Top-1) and naive averaging.

| Rectification Strategy | Text Retrieval | | Image Retrieval | | |
|---|---|---|---|---|---|
| | R@1 | R@10 | R@1 | R@10 | rSum |
| None (Discard Noise) | 70.0 | 93.9 | 50.6 | 85.5 | 467.7 |
| Hard Selection (Top-1) | 73.1 | 95.1 | 53.3 | 87.2 | 479.6 |
| Mean Pooling (Top-K) | 74.0 | 95.8 | 55.1 | 87.7 | 485.2 |
| **Graph Refiner (Ours)** | **75.1** | **97.2** | **56.5** | **89.5** | **495.2** |

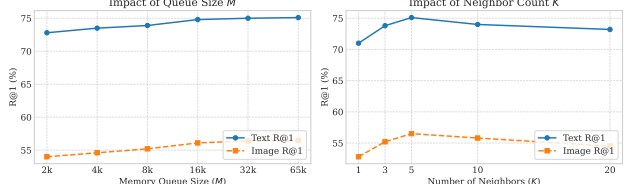

*Figure 3.* **Hyperparameter Sensitivity Analysis on R@1.** We report the R@1 performance for both Text Retrieval (Blue) and Image Retrieval (Orange). (Left) Performance improves with memory size $M$ and saturates at $M = 32k$. (Right) The retrieval accuracy peaks at $K = 5$, demonstrating that a moderate neighbor count effectively balances semantic consensus and noise introduction.

attention models the global neighborhood consensus more effectively than the fixed or local topologies of the baselines. Additionally, replacing the **Cross-Model Memory** with a "Self-Memory" variant leads to a performance drop, confirming that decoupling the query and retrieval networks is essential to mitigate confirmation bias.

*Table 5.* **Comparison of different Graph Refiner architectures** on Flickr30K ($60\%$ noise).

| Rectification Strategy | Text Retrieval | | Image Retrieval | | |
|---|---|---|---|---|---|
| | R@1 | R@10 | R@1 | R@10 | rSum |
| GCN (Fixed Graph) | 74.9 | 95.1 | 53.7 | 87.6 | 486.3 |
| GAT (Graph Attn) | 75.3 | 96.6 | 55.7 | 88.1 | 490.5 |
| **MHSA** | **75.1** | **97.2** | **56.5** | **89.5** | **495.2** |

**Hyperparameter Sensitivity Analysis.** We analyze the sensitivity of IN$^2$R to memory size $M$ and neighbor count $K$ on Flickr30K ($60\%$ noise). Figure 3 shows that performance improves with $M$ and saturates around $32k$, confirming that a sufficiently large memory captures the global distribution needed for high-fidelity retrieval. We thus adopt $M = 65,536$. Regarding $K$, performance peaks at $K = 5$. This configuration represents the optimal trade-off: it aggregates sufficient local consensus to mitigate "Single-Point Fragility" ($K = 1$) while avoiding the semantic dilution caused by distant outliers in larger neighborhoods ($K > 10$).

## 5. Conclusion

In this paper, we proposed the **Intra-modal Neighbor-aware Noise Rectification (IN$^2$R)** framework to address noisy correspondence. Departing from the limitations of dis-

crete selection, our work pioneers a shift towards **continuous prototype synthesis**, leveraging intra-modal geometric constraints to reconstruct reliable supervision targets. Extensive experiments on Flickr30K, MS-COCO, and Conceptual Captions demonstrate the superiority of our approach. IN$^2$R not only establishes new state-of-the-art results but also exhibits remarkable stability in extreme noise regimes (up to 80% noise) and real-world web-noise scenarios.

## Acknowledgments

This work was partially supported by the National Science Foundation of China under Grant 62376175, U2333211 and 22494712, the 111 Project under Grant B21044, the National Science Foundation of Sichuan Province under Grant 2025ZNSFSC0480, and the Science Fund for Creative Research Groups of Sichuan Province Natural Science Foundation under Grant 2024NSFTD0035.

## Impact Statement

This paper presents a method for training robust vision-language models using noisy datasets derived from the web. Our work contributes to the democratization of AI research by reducing the reliance on expensive, human-annotated datasets, thereby making large-scale pre-training more accessible. However, we acknowledge that our approach relies on the topological consensus of local neighborhoods to rectify noisy labels. If the underlying data distribution contains societal biases or stereotypes, reliance on local consensus could potentially amplify these biases by smoothing out minority but valid representations. While our method focuses on correcting correspondence noise, future work should consider how such rectification mechanisms interact with data fairness and bias mitigation.

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

# A. More Implementation Details

**Datasets and Features.** Following standard protocols in noisy correspondence learning (Huang et al., 2021; Han et al., 2024) we utilize pre-extracted features to ensure a fair comparison. For images, we use the bottom-up attention features extracted from a pre-trained Faster R-CNN (2048-d). For text, we use a Bi-GRU to extract sentence embeddings. These features are projected into a common $D$-dimensional metric space (e.g., $D = 1024$) via the learnable encoders $f_\theta$ and $g_\phi$. *More implementation details are provided in the supplementary material.*

**Training Settings.** Our framework is implemented in PyTorch and trained on a single NVIDIA RTX 3090 GPU. We employ the Adam optimizer with a mini-batch size of 128. The initial learning rate is set to $5e^{-4}$, with a cosine annealing decay schedule. The training process consists of two phases: a *warm-up phase* for the first $E_{warm} = 5$ epochs to initialize the feature space, followed by the *co-training phase* for another 40 epochs.

**Hyper-parameters.** For the IN$^2$R specific components:

The *Cross-Model Memory Bank size $M$* is set to 65536 for Flickr30K and 448000 for MS-COCO. For *Neighbor Retrieval*, we retrieve $K = 5$ visual neighbors to construct the semantic graph. The *Graph Refiner* is implemented as a single-layer Transformer encoder with 4 attention heads. The *Loss Weights* are empirically set as follows: the intra-modal constraint weight $\lambda_{intra} = 0.1$, and the rectification weight $\gamma = 1.0$. The *Temperature $\tau$* in the robust symmetric loss is set to 0.05, and the balancing factor $\alpha/\beta$ for SCE is set to 1.0/1.0. The margin $\alpha$ in the ranking loss is set to 0.2.

For synthetic noise experiments, we follow the standard noise generation protocol (Huang et al., 2021) to inject symmetric noise at ratios ranging from 20% to 80%.

**Training Procedure.** The framework is optimized in an end-to-end manner. In the co-training phase, the peer networks $\mathcal{M}_A$ and $\mathcal{M}_B$ are updated simultaneously but interactively. Specifically, $\mathcal{M}_A$ rectifies its noisy samples by retrieving semantic evidence from the *frozen* memory queue of $\mathcal{M}_B$ ($\mathcal{Q}_B$), and vice versa. This cross-model interaction ensures that the error flow from one network does not directly propagate to the other, strictly enforcing the decoupling principle required for robust learning.

**Inference Efficiency.** It is worth emphasizing that the proposed auxiliary modules—including the Cross-Model Memory Bank and the Graph Refiner—are exclusively constructed to facilitate robust training. **During the inference phase, these modules are discarded.** The model functions strictly as a standard dual-encoder, utilizing only the learned encoders $f_\theta(\cdot)$ and $g_\phi(\cdot)$ to compute image-text similarity. Consequently, IN$^2$R incurs **no additional computational overhead** or memory cost during deployment compared to standard baselines.

# B. Justification for Backbone Selection

*Table 6.* **Performance comparison of pure backbones** on Flickr30K (Clean setting). GPO achieves competitive performance comparable to the interaction-based SGRAF, yet maintains the efficiency of a dual-encoder architecture.

| Method | Text Retrieval | | | Image Retrieval | | |
|---|---|---|---|---|---|---|
| | R@1 | R@5 | R@10 | R@1 | R@5 | R@10 |
| GPO (Chen et al., 2021) | 74.8 | 93.5 | 97.0 | 55.1 | 83.8 | 89.4 |
| SGR (Diao et al., 2021) | 75.2 | 93.3 | 96.6 | 56.2 | 81.0 | 86.5 |
| SAF (Diao et al., 2021) | 73.7 | 93.3 | 96.3 | 56.1 | 81.5 | 88.0 |
| SGRAF (Diao et al., 2021) | 77.8 | 94.1 | 97.4 | 58.5 | 83.0 | 88.8 |

In our main experiments, we adopt the Generalized Pooling Operator (GPO) (Chen et al., 2021) as the visual-semantic backbone. This decision is driven by the inherent design philosophy of our IN$^2$R framework, which prioritizes the **"Index-and-Search"** paradigm over computationally expensive cross-modal interactions.

**Efficiency Necessity for Intra-Modal Retrieval.** While interaction-based methods like SGRAF (Diao et al., 2021) achieve superior performance on standard benchmarks (Table 6), they rely on complex graph reasoning and cross-attention mechanisms to compute similarity. This fine-grained interaction requires heavy computation for every image-text pair ($O(N^2)$ complexity), prohibiting the use of offline indexing. In contrast, our IN$^2$R framework is built upon retrieving topological neighbors from a large-scale dynamic memory bank. GPO, as a dual-encoder, maps images and texts into

compact global vectors, allowing for highly efficient nearest neighbor search via simple dot products. This efficiency is critical for the scalability of our rectification mechanism.

**Performance-Efficiency Trade-off.** As shown in Table 6, GPO delivers competitive performance comparable to SGRAF on clean datasets, serving as a strong baseline. Therefore, we select GPO to demonstrate that our performance gains stem from the proposed rectification strategy rather than a heavy backbone.

**Universality of IN$^2$R.** To further demonstrate that our method is backbone-agnostic, we integrated IN$^2$R with the SGRAF backbone on Flickr30K (60% noise). As presented in Table 7, IN$^2$R (SGRAF) consistently outperforms IN$^2$R (GPO), benefiting from the stronger representation power. However, this comes at a significant cost: the training time increases from 3.5 min/epoch to 30.0 min/epoch. We maintain that for practical large-scale retrieval, the efficiency of GPO is more valuable. Thus, we prioritize GPO in this work to highlight the efficiency and scalability of our approach.

*Table 7.* **Universality of IN$^2$R across different backbones** on Flickr30K (60% noise). While IN$^2$R boosts the performance of SGRAF, we default to GPO to balance accuracy with significantly lower training costs.

| Method | Text Retrieval | | | Image Retrieval | | | Time/Epoch |
|---|---|---|---|---|---|---|---|
| | R@1 | R@5 | R@10 | R@1 | R@5 | R@10 | |
| IN$^2$R (GPO) | 75.1 | 94.0 | 97.2 | 56.5 | 82.9 | 89.5 | **3.5 min** |
| IN$^2$R (SGRAF) | **75.8** | **94.6** | **97.3** | **57.4** | **83.1** | **89.5** | 30.0 min |

## C. More Ablation Studies

**Hyperparameter Sensitivity on Loss Weights.** Since the impact of the neighbor count $K$ and memory size $M$ has been discussed in the main text, we focus here on the sensitivity of the loss balancing terms: the **Intra-modal Constraint Weight** $\lambda_{intra}$ and the **Rectification Weight** $\gamma$. Table 8 presents the results on Flickr30K (60% noise).

*Impact of Intra-modal Weight $\lambda_{intra}$.* As shown in the left subtable, setting $\lambda_{intra} = 0$ (i.e., removing manifold stabilization) leads to a clear performance drop, verifying the necessity of geometric constraints. The performance peaks at $\lambda_{intra} = 0.5$. Increasing it further (e.g., to $1.0$) degrades the results, likely because an overly strong intra-modal constraint may dominate the optimization, interfering with the primary objective of inter-modal alignment.

*Impact of Rectification Weight $\gamma$.* Regarding $\gamma$, the model remains stable across a wide range $[0.5, 1.5]$, demonstrating that our synthesized supervision is reliable and does not require delicate tuning to balance with the clean supervision.

*Table 8.* **Sensitivity analysis of loss weights** on Flickr30K (60% noise). We analyze the trade-off between intra-modal constraints ($\lambda_{intra}$) and rectification strength ($\gamma$).

*(a)* Varying Intra-modal Weight $\lambda_{intra}$

| $\lambda_{intra}$ | Text-R@1 | Image-R@1 | rSum |
|---|---|---|---|
| 0.0 | 68.7 | 52.4 | 469.1 |
| 0.1 | 71.5 | 53.2 | 478.0 |
| **0.5** | **75.1** | **56.5** | **495.2** |
| 1.0 | 73.2 | 54.8 | 488.3 |

*(b)* Varying Rectification Weight $\gamma$

| $\gamma$ | Text-R@1 | Image-R@1 | rSum |
|---|---|---|---|
| 0.1 | 70.5 | 52.0 | 472.3 |
| 0.5 | 73.2 | 55.1 | 488.0 |
| **1.0** | **75.1** | **56.5** | **495.2** |
| 1.5 | 74.5 | 56.2 | 493.8 |

**Impact of Memory and Retrieval Mechanism.** We investigate the design of the retrieval mechanism, specifically the necessity of the **Cross-Model Decoupling** strategy. In our proposed IN$^2$R, network $\mathcal{M}_A$ retrieves neighbors from the memory queue of its peer $\mathcal{M}_B$ ($\mathcal{Q}_B$). We compare this against a *Self-Memory* baseline, where each network queries its own history ($\mathcal{Q}_A$). Table 9 demonstrates the results:

- **Self-Memory:** Relying on self-generated history leads to suboptimal performance (rSum 482.7). This suggests that without decoupling, the model tends to reinforce its own errors, leading to *confirmation bias*.

- **Cross-Memory (Ours):** By leveraging the peer network's consensus, our approach effectively breaks this self-reinforcing loop, improving rSum by 12.5 points.

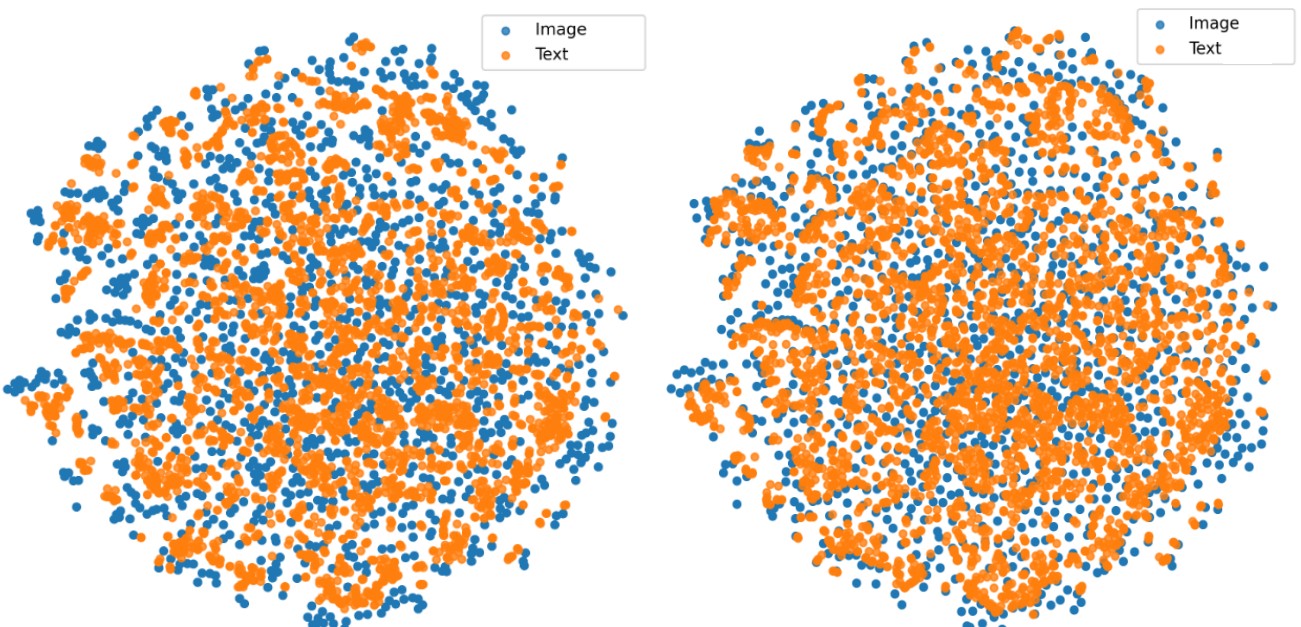

*Figure 4.* t-SNE visualization of the learned feature embeddings on the Flickr30K test set. (Left) Our proposed IN$^2$R: The feature distribution exhibits a more compact and structured manifold, indicating that our intra-modal geometric constraints successfully stabilized the feature space. (Right) Discrete Selection Baseline: The feature space appears more scattered and disordered. Compared to the discrete selection paradigm, IN$^2$R achieves better visual-semantic alignment, where image (blue) and text (orange) features of similar semantics form tighter clusters.

- **Queue Strategy:** We also validate the *Elitist Rolling Update* (filtering low-confidence samples). Removing this filter (i.e., storing all samples) drops performance to 488.1, confirming that maintaining a high-confidence "elite" memory is vital for pristine retrieval.

*Table 9.* **Ablation of Memory Construction and Retrieval Strategy** on Flickr30K (60% noise). "Decoupling" indicates whether peer-memory is used.

| Memory Strategy | Decoupling | Text R@1 | Image R@1 | rSum |
|---|---|---|---|---|
| Self-Memory (Standard) | | 72.5 | 53.8 | 482.7 |
| Full Queue (No Filtering) | ✓ | 73.4 | 54.9 | 488.1 |
| **Elitist Cross-Memory (Ours)** | ✓ | **75.1** | **56.5** | **495.2** |

### C.1. Qualitative Analysis of Feature Manifolds.

To intuitively verify the impact of our proposed method on the feature space, we visualize the learned image and text embeddings on the Flickr30K test set using t-SNE. Figure 4 provides a side-by-side comparison between our IN$^2$R (Left) and the baseline trained with discrete selection (Right). As observed on the right side of Figure 4, the feature space learned by the discrete selection paradigm appears relatively dispersed, with blurred boundaries between semantic clusters. This visual scattering corroborates our hypothesis that assigning discrete, noisy proxies introduces discretization error, preventing the model from learning a sharp semantic structure. In contrast, the manifold learned by IN$^2$R on the left side exhibits significantly higher intra-class compactness and inter-class separability. The image (blue) and text (orange) features form tighter, more distinct clusters, indicating a superior cross-modal alignment. This structural improvement demonstrates that our Graph-Guided Continuous Rectification effectively filters out feature-level noise and synthesizes reliable supervision targets, thereby regularizing the manifold towards a more discriminative geometric structure.

