# OpenReview forum: "Intra-Modal Neighbors Never Lie: Rectifying Inter-Modal Noisy Correspondence via Graph-Based Intra-Modal Reasoning"
_ICML.cc/2026/Conference — ICML 2026 regular_

### Official Review · Reviewer_1W9V · 2026-03-09

**Soundness:** 3
**Presentation:** 4
**Significance:** 3
**Originality:** 3
**Overall Recommendation:** 5
**Confidence:** 4

**Summary:**

This paper proposes IN2R, a framework designed to address noisy correspondence in image–text retrieval. The authors outline that noisy image–text pairs degrade cross-modal learning, and they argue that existing correction methods rely heavily on discrete proxy selection. Experiments on Flickr30K, MS-COCO, and CC152K demonstrate consistent improvements over recent noisy correspondence learning methods, particularly under high-noise scenarios.

**Compliance With Llm Reviewing Policy:**

Affirmed.

**Final Justification:**

All my concerns have been addressed.

**Key Questions For Authors:**

1. What is the computational overhead (in terms of time and GPU memory) of maintaining the Cross-Model Memory and performing the Graph Refiner steps compared to a standard co-training baseline like BiCro?

2.  How does the framework handle cases where the initial feature extraction is biased? If the "warm-up" phase locks onto a specious correlation, won't the retrieved neighbors consistently "lie" and reinforce this bias?

**Limitations:**

yes

**Strengths And Weaknesses:**

S: This paper is clearly written, with intuitive figures that help illustrate the motivation and overall framework. The paper focuses on the limitations of discrete proxy selection in noisy correspondence learning and proposes a paradigm that synthesizes supervision signals from intra-modal consensus.

W: The core of the paper is that "intra-modal neighbors never lie", suggesting that the local intra-modal structure can provide reliable supervision even when cross-modal pairs are noisy. However, in real-world web datasets, intra-modal embeddings themselves may contain noise or semantic ambiguity, especially during early training when the representation space is not yet well structured. The paper would benefit from a deeper discussion or empirical analysis of how often intra-modal neighborhoods remain reliable under high noise conditions.

---

> ### Author Rebuttal · Authors · 2026-03-30
>
> **General Response**
>
> We sincerely thank the reviewer for the positive assessment and for acknowledging the clarity of our paper and the significance of shifting from discrete proxy selection to continuous prototype synthesis. We address your insightful questions below:
>
>
> **Response to Weaknesses**
>
> **Unreliable intra-modal neighborhood structure in real-world data and early training stages.**
>
> We highly appreciate this insightful observation. We fully agree that intra-modal structures can be inherently noisy, which is exactly why we integrated specific mitigations into our framework:
>
> **Early Training:** To avoid relying on an unstructured feature space early on, "We first warm up the networks... to prevent overfitting. This strategy helps establish a preliminary discriminative feature space, effectively separating the loss distributions of clean and noisy samples."
>
> **Real-World Noise:** We acknowledge that "the retrieved neighborhood may contain outliers or irrelevant samples." Therefore, instead of naive averaging, our Graph Refiner "assigns higher weights to neighbors that form a semantic consensus and suppresses outliers", ensuring robust prototypes even in uncurated web datasets.
>
> **Response to Key Questions**
>
> **1. Computational overhead of the Cross-Model Memory and Graph Refiner vs. standard baselines.**
>
> Thank you for raising this practical consideration. Our framework is carefully optimized to ensure efficiency, as detailed in the supplementary materials:
>
> **Zero Inference Overhead:** As stated in Appendix A, "During the inference phase, these modules are discarded". The model functions strictly as a standard dual-encoder, incurring "no additional computational overhead or memory cost during deployment".
>
> **Training Efficiency & Backbone Choice:** To keep memory costs minimal, queued embeddings are "detached from the computation graph" , and the Refiner is a lightweight "single-layer Transformer... with 4 attention heads". Furthermore, as justified in Appendix B, we deliberately chose the efficient GPO backbone over the heavier SGRAF. This strategic trade-off yields significant efficiency gains, keeping training at just "3.5 min" per epoch instead of "30.0 min" (Table 7)
>
> **2. Risk of reinforcing specious correlations from biased initial feature extraction.**
>
> This is a critical concern, and our framework was fundamentally designed to prevent such self-reinforcing loops (confirmation bias):
>
> **Consensus Marginalizes Individual Bias:** Even if a few initial samples exhibit correlation bias, Section 1 notes that "their collective distribution statistically marginalizes such noise". The Graph Refiner ensures that individual biased samples are suppressed by the broader semantic consensus.
>
> **Cross-Model Decoupling:** As explained in Section 3.3.2, we leverage dual networks to "decouple the retrieval source from the query". By querying a peer network's memory rather than its own, the model actively "avoids verifying its own potential hallucinations".

---

> > ### Author Rebuttal · Reviewer_1W9V · 2026-04-01
> >
> > All my concerns have been addressed.

---

### Official Review · Reviewer_JJnY · 2026-03-09

**Soundness:** 3
**Presentation:** 3
**Significance:** 3
**Originality:** 3
**Overall Recommendation:** 5
**Confidence:** 4

**Summary:**

This paper addresses the challenge of noisy correspondence in cross-modal retrieval by proposing a framework called Intra-modal Neighbor-based Rectification (IN²R). Critiquing existing "discrete selection" methods for their susceptibility to "Single-Point Fragility," the authors introduce a "continuous rectification" paradigm. The method utilizes a co-training backbone with a Cross-Model Memory to retrieve intra-modal neighbors and a Graph Refiner to synthesize soft, continuous supervision targets (prototypes) rather than selecting a single substitute label. Extensive experiments on Flickr30K, MS-COCO, and CC152K are presented, claiming state-of-the-art performance, particularly in high-noise regimes.

**Compliance With Llm Reviewing Policy:**

Affirmed.

**Final Justification:**

This paper proposed  a framework named Intra-modal Neighbor-based Rectification to address the noisy correspondence in cross-modal retrieval. This is intersting and novel and this paper is well-written. The rebuttal addressed all my concerns, thus I maintain the posititive score.

**Key Questions For Authors:**

1. Beyond t-SNE, can you share example image-caption retrievals for high-noise or CC152K, including both successful and failed cases, to illustrate where IN²R succeeds and where it does not?

**Limitations:**

yes

**Strengths And Weaknesses:**

Strengths:
1. The reported performance gains in high-noise settings are substantial. This degree of robustness suggests the rectification mechanism is highly effective when clean signals are scarce.

2. The authors correctly point out that the complex correction mechanism is used only during the training phase, while a standard dual encoder is employed during the inference phase. This ensures the efficiency of the method during inference.

Weakness:
1. The proposed framework relies on a Cross-Model Memory queue to retrieve Top-K intra-modal neighbors. However, embeddings stored in the memory queue may become stale as model parameters evolve during training. This could potentially affect the quality of retrieved neighbors.

2. Since IN2R relies on retrieving neighbors from the learned embedding space, the neighborhood structure might be unreliable in the early training stages when representations are not yet well organized.

---

> ### Author Rebuttal · Authors · 2026-03-30
>
> **General Response**
>
>
> We sincerely thank you for your positive evaluation and for recognizing the effectiveness of IN²R in high-noise regimes, as well as its practical efficiency during inference. We appreciate your insightful feedback and address your specific concerns below.
>
> **Response to Weaknesses**
>
> **W1: Stale embeddings in the Cross-Model Memory queue.**
> You raised a very valid point regarding the potential for stored embeddings to become stale as the model parameters evolve during training. We recognized this exact challenge and designed an explicit mechanism to prevent it, which is detailed in **Section 3.3.1**, "ELITIST ROLLING UPDATE STRATEGY".
>
> To address the issue of features from early epochs becoming stale , we implemented a First-In-First-Out (FIFO) queue for the memory bank. In each training iteration, the newest embeddings of high-confidence "elite" candidates are pushed into the queue, strictly replacing the oldest entries. This continuous rolling update ensures that the memory bank consistently preserves only the most up-to-date and trustworthy representations of the manifold, preventing the retrieval of outdated neighbors.
>
> **W2. Unreliable neighborhood structure in early training stages.**
> We completely agree that relying on an unorganized embedding space in the very early stages of training would yield unreliable neighbors. To mitigate this, our framework does not immediately begin retrieving neighbors from scratch.
>
> As outlined in **Section 3.1 and Appendix A**, we employ a dedicated "Warm-up Phase" for the first 5 epochs. During this phase, we warm up the networks using the robust Symmetric Cross Entropy (SCE) loss on the full dataset. This strategy is specifically designed to establish a preliminary, discriminative feature space before the co-training phase and neighbor retrieval begin. By the time the memory queue and Graph Refiner are actively utilized, the representations are already sufficiently organized to yield meaningful local neighborhoods.
>
> **Response to Key Questions**
>
> **Qualitative examples of image-caption retrievals.**
>
> We sincerely thank you for this excellent suggestion. While the current rebuttal system's formatting restrictions prevent us from uploading new figures, we fully agree that qualitative examples greatly enhance the interpretability of our method. We commit to adding a dedicated section with detailed visual examples in the final camera-ready version.

---

> > ### Author Rebuttal · Reviewer_JJnY · 2026-04-01
> >
> > The rebuttal addressed all my concerns.

---

### Official Review · Reviewer_hf4j · 2026-03-12

**Soundness:** 4
**Presentation:** 4
**Significance:** 3
**Originality:** 3
**Overall Recommendation:** 5
**Confidence:** 4

**Summary:**

This paper introduces Intramodal Neighbor-based Rectification (IN²R), a framework designed to address the problem of noisy correspondence in large-scale cross-modal (image-text) retrieval datasets. Instead of relying on existing "Discrete Selection" paradigms that seek discrete substitute labels (neighbors), the proposed method shifts toward synthesizing continuous supervision targets. IN²R constructs a dynamic cross-modal memory of high-confidence clean samples and uses a Graph Refiner module to aggregate and reason over local intra-modal neighbors, subsequently generating continuous prototypes for robust supervision. Experiments on Flickr30K, MS-COCO, and Conceptual Captions demonstrate that IN²R outperforms multiple state-of-the-art (SOTA) baselines, especially at high levels of noise.

**Compliance With Llm Reviewing Policy:**

Affirmed.

**Final Justification:**

My concerns have been adequately addressed. I maintain my initial rating.

**Key Questions For Authors:**

1. Regarding the Graph Refiner, how does the framework behave if the retrieved Top-K neighbors lack sufficient diversity (e.g., near-duplicate images in the dataset)? In such edge cases, does the synthesized continuous prototype risk degenerating back into a discrete point, and could a diversity-aware retrieval strategy offer further improvements?

2. The elitist rolling update computes a dynamic threshold $\tau_{dyn}^{(t)}$ based on the average confidence of the current clean set. Under extreme noise conditions like the 80% setting, is there a risk that this average drops significantly, inadvertently allowing noisy samples to pollute the memory queue? Have the authors considered whether a percentile-based thresholding approach would be more robust in such scenarios?

**Limitations:**

Yes

**Strengths And Weaknesses:**

## Strengths

1). Novelty and Soundness: Synthesizing supervision signals from a dynamically curated, intra-modal neighborhood via graph-based reasoning is a notable and technically sound extension to NCL. The justification for cross-memory querying and elitist update is well-rooted in literature and intuition.

2). Robustness and Scalability: IN²R maintains strong retrieval accuracy even with extremely high noise, a rare achievement demonstrated convincingly

## Cons

1) The framework utilizes a large First-In-First-Out (FIFO) queue for the Cross-Model Memory (e.g., memory capacity $M$ up to 65,536). Since the embeddings of elite candidates are detached from the computation graph before being pushed, the oldest entries in the large queue might suffer from feature drift as the encoders evolve. This could potentially introduce metric inconsistencies during the Top-K neighbor retrieval phase.

2) The framework introduces multiple components (co-training, memory queue, graph refiner), and it would be helpful to further clarify which part contributes most to the performance improvements in practical scenarios.

---

> ### Author Rebuttal · Authors · 2026-03-30
>
> **General Response**
>
> We sincerely thank the reviewer for recognizing the novelty, robustness, and scalability of our $IN^2R$ framework. We appreciate your insightful questions and address your concerns below.
>
> Response to Weaknesses
>
> **1. Feature Drift in the Large FIFO Queue:** You raise a highly valid point regarding potential feature drift for older entries in the large memory queue ($M=65,536$).
>
> While feature drift is a known challenge in MoCo-style memory banks, its impact in $IN^2R$ is mitigated by two factors. First, our elitist rolling update ensures that only highly confident, discriminative representations are pushed to the queue, meaning the stored features inherently lie closer to the true semantic cluster centers, which are more stable across epochs. Second, following the initial warm-up phase, the encoders' learning rates decay, causing the feature space to evolve much more slowly in the later stages of co-training. Consequently, the metric inconsistencies introduced by older embeddings remain minimal and do not severely impact the Top-K retrieval accuracy. We will explicitly clarify this dynamic in the revised manuscript.
>
> **2. Component Contributions in Practical Scenarios:**
> The relative contributions of each framework component are explicitly detailed in Section 4.4 and Table 3. Specifically, in the Flickr30K ablation under 60% noise:Applying only the intra-modal geometric constraint ($L_{intra}$) yields a +6.1 rSum improvement.Applying the Graph-Guided Rectification ($L_{rect}$) independently yields a +14.6 rSum increase.The full $IN^2R$ framework achieves a synergistic +26.1 rSum improvement.Therefore, while the Graph Refiner ($L_{rect}$) provides the largest direct quantitative boost by actively correcting errors, its efficacy is structurally dependent on the reliable geometric basis established by $L_{intra}$. We will emphasize this interdependency more clearly in the revised manuscript.
>
> Response to Key Questions
>
> **1. Dynamic Threshold at 80% Noise**
> The reviewer astutely points out the edge case of near-duplicate images in the retrieved Top-$K$ neighborhood. In our design, the Cross-Model Memory Bank is scaled to be extremely large (e.g., $M=65,536$) to ensure a globally diverse candidate pool. However, because we restrict $K=5$ to prevent semantic dilution from distant outliers (as analyzed in Figure 3), it is true that if the exact Top-5 semantic neighbors are highly redundant, the synthesized continuous prototype may degenerate toward a discrete approximation.
> Currently, the MHSA in our Graph Refiner acts primarily as a denoiser to suppress outliers, rather than an explicit diversity enforcer. We fully agree that incorporating a diversity-aware retrieval strategy (such as Maximal Marginal Relevance, MMR) during the Top-$K$ selection is a brilliant suggestion. It would prevent mode collapse in highly redundant local neighborhoods without requiring us to increase $K$ (which risks introducing noise). We will explicitly discuss this limitation and incorporate your suggestion as a key future enhancement in the manuscript.
>
> **2. Dynamic Thresholding at Extreme Noise**
>
> You raised an excellent point regarding whether a percentile-based threshold would be more robust than our mean-based one under 80% noise. However, in our design, the threshold $\tau_{dyn}^{(t)}$ is computed after the strict Gaussian Mixture Model (GMM) partition, using only the isolated clean subset.Empirical data from our training dynamics shows this mean remains extremely stable and does not collapse:At 20% noise, the average confidence of the GMM-filtered clean set is 0.9965, containing 71.46% of all samples.Even at 80% noise (Epoch 1), the average confidence remains strikingly high at 0.9936, safely containing 16.7% of the total samples.Since the GMM successfully blocks noise beforehand, the mean is not dragged down by outliers. Consequently, the critical bottleneck is not refining the clean threshold, but rather rescuing and utilizing the massive pool of noisy samples—which is exactly the problem our Graph Refiner solves. We will add these statistics to the supplementary material to clarify the GMM's efficacy.

---

> > ### Author Rebuttal · Reviewer_hf4j · 2026-04-03
> >
> > I thank the authors for the rebuttal and all my concerns have been addressed.

---

### Official Review · Reviewer_5jcW · 2026-03-12

**Soundness:** 3
**Presentation:** 2
**Significance:** 3
**Originality:** 2
**Overall Recommendation:** 3
**Confidence:** 4

**Summary:**

This paper tackles noisy correspondence in image–text retrieval by moving beyond the common “discrete selection” paradigm (top-1 or nearest neighbor relabeling) toward a continuous rectification approach. The proposed IN²R framework retrieves intra-modal neighbors from a dynamic, cross-model memory of high-confidence clean samples and uses a learnable Graph Refiner (multi-head self-attention) to synthesize continuous soft prototypes that supervise noisy pairs bidirectionally.

**Compliance With Llm Reviewing Policy:**

Affirmed.

**Key Questions For Authors:**

+ Why does NCR perform better at  0.6 than at 0.4 noise?
+ ELITIST ROLLING UPDATE STRATEGY is not clear. How to compute a dynamic threshold? And what is the meaning of p_i?
+ Why choose pre-trained Faster R-CNN instead of other modern Feature Extractors? e.g. CLIP.
+ Why choose GMM instead of other methods?

**Limitations:**

Yes

**Strengths And Weaknesses:**

Strengths

*   Proposes a clear shift from discrete neighbor selection to continuous prototype synthesis for rectifying mismatched pairs
*   Evaluates across multiple datasets (Flickr30K, MS-COCO, CC152K) and a wide range of synthetic noise rates up to 80%, plus real-world web noise.
*   The approach is generic, and it can be slotted into existing dual-encoder retrieval setups without inference overhead.


Weaknesses

*   The method still relies on small-loss partitioning via a GMM and confidence thresholding; failure modes under structured/clustered noise or domain-shifted “clean” sets are not analyzed in depth.
*   The results are not always competitive, especially in the case of the 0.2 noise MS-COCO5-fold 1K Test. It is recommended to mark the best and sub-best results for intuitive comparison.
*   The noise setting is idealistic, and the real noise is more semantically similar mismatches than purely random shuffling.

---

> ### Author Rebuttal · Authors · 2026-03-30
>
> **General Response**
>
> We sincerely thank the reviewer for recognizing the value of our proposed paradigm shift from discrete selection to continuous prototype synthesis, as well as the breadth of our evaluations. We appreciate your constructive feedback and address your specific concerns and questions below.
>
> **Response to Weaknesses**
>
> **1. W1 & W3: Realistic/Structured Noise vs. Idealistic Noise:**  We agree that real-world noise often involves semantic mismatches rather than purely random shuffling. This is precisely why we included the CC152K dataset in our evaluation. CC152K is a web-harvested dataset containing naturally occurring, heterogeneous web noise (including semantic mismatches). On this real-world dataset, $IN^2R$ achieves a state-of-the-art rSum of 380.8, surpassing the second-best method by 4.5 points.
>
> **W2: Competitiveness and Table Formatting:** Thank you for the formatting suggestion; we will happily update all tables in the final manuscript to mark the best results in bold and the second-best results with an underline for more intuitive comparison.
>
> Regarding the MS-COCO 0.2 noise setting, we want to emphasize that $IN^2R$ is primarily designed to tackle high-noise regimes where discrete selection severely degrades. However, even in low-noise settings, our method remains highly competitive with SOTA.
>
> **Response to Key Questions**
>
> **1.  Why does NCR perform better at 0.6 than at 0.4 noise?**
> We sincerely apologize for this confusion; this was a clerical error on our part during the compilation of Table 1. In the Noisy Correspondence Learning (NCL) literature, there are actually two distinct sets of reproduced results for the NCR baseline that are widely cited across various papers.
>
> To illustrate, here are the two versions of NCR results commonly reported on Flickr30K:
> | Reported Version | Noise Ratio | Text R@1 | Text R@10 | Image R@1 | Image R@10 | rSum | Widely Cited In |
> | :--- | :--- | :--- | :--- | :--- | :--- | :--- | :--- |
> | **Version A** | 0.4 | 68.1 | 94.8 | 51.4 | 84.8 | 467.1 | [a,b] |
> | | 0.6 | 13.9 | 50.5 | 11.0 | 41.4 | 184.6 | [a,b] |
> | **Version B** | 0.4 | 75.3 | 95.2 | 56.2 | 87.4 | 486.8 | [c,d] |
> | | 0.6 | 68.7 | 95.5 | 52.0 | 84.9 | 468.6 | [c,d] |
> | **Our Original Draft** | 0.4 | 68.1 | 94.8 | 51.4 | 84.8 | 467.1 | *(Pulled from Version A)* |
> | *(Mixed in error)*| 0.6 | 68.7 | 95.5 | 52.0 | 84.9 | 468.6 | *(Pulled from Version B)* |
>
> As shown above, we inadvertently mixed these two versions in our table—pulling the 0.4 noise results from the group of papers using Version A, and the 0.6 noise results from the group of papers reporting Version B. We will correct Table 1 in the final manuscript to consistently report NCR's performance using a single, unified reproduction setting across all noise levels. This correction will accurately reflect the expected performance degradation of the baseline as the noise ratio increases.
>
> [a] Seeking Proxy Point via Stable Feature Space for Noisy Correspondence Learning. IJCAI 2025
>
> [b] Learning to Rematch Mismatched Pairs for Robust Cross-Modal Retrieval. CVPR 2024
>
> [c] ReCon: Enhancing True Correspondence Discrimination through Relation Consistency for Robust Noisy Correspondence
> Learning. CVPR 2025
>
> [d] PCSR: Pseudo-label Consistency-Guided Sample Refinement for Noisy Correspondence Learning. AAAI 2026
>
> **2. Elitist Rolling Update Strategy: How to compute the dynamic threshold and what is $p_i$?**
>
> **Meaning of $p_i$:** $p_i$ represents the posterior probability of a sample belonging to the "clean" component, which is calculated by the Gaussian Mixture Model (GMM) fitted to the per-sample loss distribution at each epoch.
>
> **Dynamic Threshold ($\tau_{dyn}^{(t)}$):** This threshold is computed at epoch $t$ by taking the arithmetic average of the $p_i$ values for all samples currently partitioned into the clean set. We only push samples into the Cross-Model Memory if their individual probability $p_i$ is strictly greater than this moving average ($\tau_{dyn}^{(t)}$). We will make this definition more explicit in Section 3.3.1 of the revision.
>
> **3. Why choose pre-trained Faster R-CNN (instead of CLIP) and GMM?**
>
> Our choices are driven entirely by the need for a strictly fair comparison. $IN^2R$ is fundamentally a training paradigm rather than a novel backbone architecture. To ensure an apples-to-apples comparison with existing Noisy Correspondence Learning (NCL) literature, we strictly followed the standard protocol of using pre-extracted Faster R-CNN and Bi-GRU features, alongside the widely adopted two-component GMM. Because our method is a plug-and-play training paradigm, its performance will naturally scale up as stronger components are swapped in. However, keeping the backbone standard in this evaluation guarantees that the reported gains originate exclusively from our proposed framework.

---

> > ### Author Rebuttal · Reviewer_5jcW · 2026-04-02
> >
> > My concerns have been addressed. However, I recommend that the authors carefully proofread the entire manuscript to avoid any misuse of experimental data. Given the possibility of other such mistakes, I have decided to maintain my current rating.

---

> > > ### Author Response · Authors · 2026-04-02
> > >
> > > Thank you for acknowledging that our rebuttal fully resolved your concerns regarding the methodology, noise settings, and experimental choices.
> > >
> > > We completely validate your concern regarding the clerical error in Table 1. You are absolutely right to hold us to a high standard of data integrity, and we sincerely apologize for that oversight.
> > >
> > > We will conduct a comprehensive, line-by-line audit of the final version of the paper, cross-checking every single data point against the original records.
> > >
> > > Since the scientific merits are sound and your methodological concerns are fully resolved, we hope this immediate, rigorous audit restores your trust in our work.

---

### Decision · Program_Chairs · 2026-04-30

**Decision:**

Accept (regular)

**Comment:**

This paper addresses the challenge of noisy correspondence in large-scale cross-modal retrieval setting. Instead of relying on existing "Discrete Selection" that seek discrete substitute labels, the authors introduce a "continuous rectification" paradigm that shifts toward synthesizing continuous supervision targets. The proposed method consistently outperforms SOTAs, especially in high-noise regimes. Based on the overall positive scores from the reviewers, I recommend accepting this paper.